# Multi-objective Bayesian active learning for MeV-ultrafast electron diffraction

Fuhao Ji [1] ✉, Auralee Edelen [1], Ryan Roussel[1], Xiaozhe Shen [1],
Sara Miskovich [1], Stephen Weathersby[1], Duan Luo[1], Mianzhen Mo [1],
Patrick Kramer[1], Christopher Mayes [1], Mohamed A. K. Othman[1], Emilio Nanni [1],
Xijie Wang [1], Alexander Reid [1], Michael Minitti [1] & Robert Joel England [1] ✉

Ultrafast electron diffraction using MeV energy beams(MeV-UED) has enabled unprecedented scientific opportunities in the study of ultrafast structural dynamics in a variety of gas, liquid and solid state systems. Broad scientific applications usually pose different requirements for electron probe properties. Due to the complex, nonlinear and correlated nature of accelerator systems, electron beam property optimization is a time-taking process and often relies on extensive hand-tuning by experienced human operators. Algorithm based efficient online tuning strategies are highly desired. Here, we demonstrate multi-objective Bayesian active learning for speeding up online beam tuning at the SLAC MeV-UED facility. The multi-objective Bayesian optimization algorithm was used for efficiently searching the parameter space and mapping out the Pareto Fronts which give the trade-offs between key beam properties. Such scheme enables an unprecedented overview of the global behavior of the experimental system and takes a significantly smaller number of measurements compared with traditional methods such as a grid scan. This methodology can be applied in other experimental scenarios that require simultaneously optimizing multiple objectives by explorations in high dimensional, nonlinear and correlated systems.

Ultrafast electron diffraction using MeV energy beams (MeV-UED) has led to a new paradigm in ultrafast electron scattering[1,2]. MeV electron probes lead to atomic spatial resolution together with sub-picosecond temporal resolution[3]. Furthermore, MeV electrons can provide pure kinetic scattering signals for samples with a given thickness, which makes it possible to achieve quantitatively understood observables to validate physical modeling. Recently, MeV-UED techniques have reached a high level of maturity with several beamlines operating as user facilities. Rapid advancements in MeV fs electron probes, sample delivery and manipulation techniques, high efficiency electron detection and novel data analysis techniques[3–7] have enabled unprecedented scientific opportunities in ultrafast structural dynamics, including observations of bond dissociation, coherent ground state wavepacket motion after electrocyclic ring-opening[8], simultaneous observation of electronic and nuclear structure changes in a molecule undergoing non-adiabatic dynamics[9], hydrogen bond strengthening in liquid water[10], and phase switches of materials with exotic functional properties[11,12].

Broad scientific applications usually pose different requirements for electron beam properties, such as temporal length, spot size and transverse momentum resolution in reciprocal space (q-resolution). Measuring these beam properties as a function of the input parameters using limited and time-consuming diagnostics becomes a necessary part of planning and optimization of experiments. Due to the complex, nonlinear and correlated nature of accelerator systems, electron beam property optimization often relies on extensive and time-taking hand-tuning by experienced human operators. This comes at the cost of reducing the beam time available for scientific experiments. Uniformly

[1]SLAC National Accelerator Laboratory, Menlo Park, 94025 California, USA. ✉e-mail: fuhaoji@slac.stanford.edu; england@slac.stanford.edu

spaced grid-like parameter scans are routinely conducted in one or two dimensions for sampling the parameter space and obtaining an overview of the beam property behaviors. However, grid scans suffer from poor scaling as the number of sampling points increases exponentially with input parameter dimensions which may be impractical during a limited beam time. Algorithm based efficient online tuning strategies are highly desired as they hold the potential to reducing the setup time, allowing human experts to focus on more challenging high-level scientific problems.

Recently, Artificial intelligence and Machine learning (AI/ML) have reached sufficient maturity for applications in real-world tasks such as autonomous driving, natural language processing[13], protein folding prediction[14] and fusion reactor control[15]. Speeding up and aiding online optimizations of complex particle accelerators is one of the key areas where AI/ML can make substantial contributions[16–23]. Bayesian Optimization (BO) with Gaussian processes(GPs)[24,25], often referred to as Bayesian active learning[22,26,27], represents a class of ML algorithms in which a probabilistic model and its uncertainty at each measurement step are used to select the optimal next observation point. Recently, BO based optimization methods have been successfully applied for online tuning at large-scale accelerator systems such as LCLS[19], with a lower number of observations needed than Gradient-based or Nelder-Mead simplex algorithms. These algorithms are designed to optimize a single beam property (an "Objective"). In practice, beam tuning often seeks to simultaneously optimize multiple Objectives. This pose a significant challenge as individual beam properties may not be independent optimizables due to complex inter-correlations between them. For example, in UED it is difficult to simultaneously minimize the electron pulse length and spot size/q-resolution due to Space Charge (SC) forces. The goal of multi-objective optimization is to determine the Pareto Front (PF) which represents the system performance limit and gives the trade-offs between key objectives. In this domain, some of the most popular methods are the evolutionary algorithms for their simplicity in implementation and broad applicability[28]. However, the evolutionary algorithms are extremely data inefficient as they require evaluation of large number of candidate solutions to converge to PF. As a result, the evolutionary algorithms are nearly impossible to use for online accelerator tunings[23]. To this end, the Multi-Objective Bayesian Optimization (MOBO) scheme[20,29] was recently introduced. This method uses a set of GP surrogate models, along with a multi-objective HyperVolume (HV) improvement

acquisition function, to reduce the number of observations needed. This method can converge by at least an order of magnitude more quickly than evolutionary algorithms, such as Multi-Objective Genetic Optimization (MOGA), and is thus a critical step toward online multi-objective optimization on real accelerator systems.

In this work, we demonstrate Multi-Objective Bayesian Active Learning for speeding up online beam tuning on an MeV-UED facility at SLAC National Accelerator Laboratory. The MOBO algorithm was used for searching the parameter space efficiently in a serialized manner, and actively learning the PF giving trade-offs of key beam properties (electron pulse length, q-resolution and spot size at sample) that have a direct impact on the outcome and quality of scientific user experiments. The achieved performance was comparable with that obtained by experienced human operators. The MOBO scheme enables an unprecedented overview in the multi-dimensional parameter space with a significantly smaller number of measurements required compared with traditional exploration methods such as a Grid Search (GS) scan. While our focus here is the application of MOBO in beam tuning for MeV-UED, the explained methodology can be applied in other experimental scenarios involving high dimensional, nonlinear and correlated systems.

## Results

We conducted experiments at the SLAC MeV-UED facility to demonstrate the feasibility of applying MOBO for online exploration and optimizations of electron beam properties in a multi-dimensional parameter space. The experimental setup and MOBO workflow are illustrated in Fig. 1. The MeV-UED beamline generates 4.2 MeV energy electron pulses using an S-band photoinjector triggered by 60 fs FWHM laser pulses. The electron pulse length downstream is sensitive to the launch phase of the electrons relative to the photoinjector RF field (referred to as the "gun phase" and characterized by the parameter $\phi$). The electron pulses are then focused using two magnetic solenoids. One is referred to as the "gun solenoid" and characterized by the parameter $B_1$, the other is referred to as the "micro-focus solenoid" and characterized by the parameter $B_2$. The electrons are delivered to the Interaction Point (IP) for performing pump-probe electron diffraction experiments. The electron diffraction patterns are then recorded and used to extract fs time scale structural dynamics signals. Here, three electron beam properties are optimized. The electron pulse length ($\sigma_t$) at IP sets the smallest temporal resolution that can be

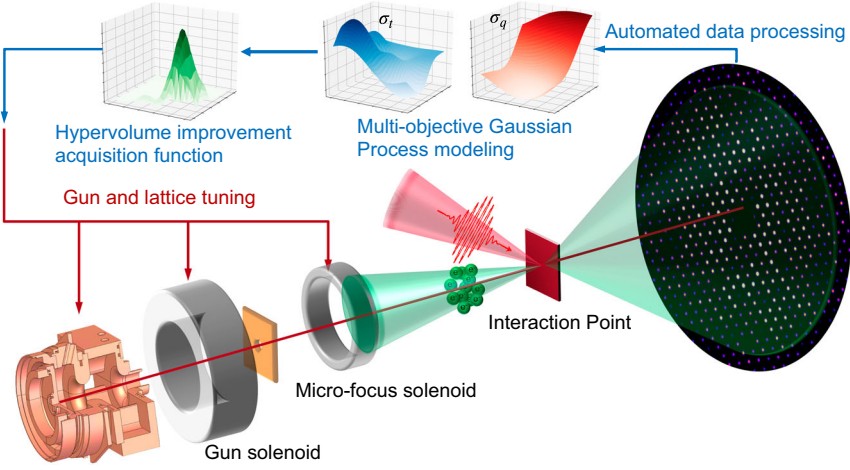

**Fig. 1 | Cartoon depicting the Multi-Objective Bayesian active Learning experiment at SLAC MeV-UED.** Ultrafast electron pulses are generated using an S-band photoinjector and delivered to interaction point for conducting fs electron diffraction experiments. Electron beam properties are extracted from detector readouts and used for constructing GP surrogate models giving overview of machine response in the parameter space. A hypervolume improvement acquisition function is constructed for predicting the optimal next observation point and efficiently search the parameter space for obtaining PFs.

achieved for resolving fast dynamical processes. The spot size ($\sigma_x$) at IP sets the transverse probing area and the q-resolution ($\sigma_q$) on the diffraction detector represents the transverse momentum resolution in the reciprocal space. Due to space charge forces, $\sigma_t$, $\sigma_x$ and $\sigma_q$ are competing properties that can not be simultaneously minimized. We used MOBO to explored the response of $\sigma_t$, $\sigma_x$ and $\sigma_q$ to the input beamline parameters [$\phi$, $B_1$, $B_2$] and obtained the PFs giving trade-offs between them. In each iteration, MOBO evaluates a candidate point in the parameter space and updates the GP surrogate model with the newly acquired data. The Expected Hypervolume Improvement (EHVI)[30] acquisition function is then used to calculate the optimal next observation point (see methods). The process is performed iteratively and ultimately providing an approximation of the best achievable PF.

We first conducted $\sigma_t$ vs $\sigma_q$ optimizations by varying gun phase ($\phi$) and solenoid strengths ($B_1$ and $B_2$). The initial pulse charge was set to be 10 fC. The gun amplitude was 90 MV/m. A 200 $\mu$m collimator was used as an emittance filter located after the gun solenoid. We initialized the exploration with 10 random sampling points in the chosen subdomain of input parameter space (Table 1). Fig. 2 illustrates the exploration result after 121 measurements. For each measurement, three observations of $\sigma_t$ and $\sigma_q$ were recorded for averaging the fluctuations due to system jitters. These measurements are then used

to train GP models of $\sigma_t$ and $\sigma_q$ responses to the input parameter $\phi$, $B_1$ and $B_2$. A Matérn kernel with $\nu = 5/2$ was used and hyperparameter training was done by maximizing the log marginal likelihood of the GP model with the gradient based Adam optimization algorithm. Measurements of $\sigma_t$ projected to the $\phi$ - $B_2$ subspace, along with the posterior mean predicted by the GP model, are plotted in Fig. 2a. Estimated length scales from the GP models are shown in Table 1. The $\mathcal{L}(\sigma_t)$ for $\phi$, $B_1$ and $B_2$ are at the same scale, implying that $\sigma_t$ depends on all three input parameters. This is consistent with previous observations as $\sigma_t$ depends on both the temporal-energy chirp at gun exit (changed by $\phi$) and SC induced temporal broadening during beam transport (changed by $B_1$ and $B_2$). Fig. 2b, c illustrates the $\sigma_t$ and $\sigma_q$ responses projected to the $B_1$ - $B_2$ subspace. The $\mathcal{L}(\sigma_q)$ for $B_1$ is significantly shorter than that for $B_2$, indicating that $\sigma_q$ varies faster with $B_1$. Again, this is consistent with previous observations that q-resolution mainly depends on the transmitted beam emittance and gun solenoid strength. On the other hand, The $\mathcal{L}(\sigma_q)$ for $\phi$ is significantly larger than the $\mathcal{L}(\sigma_q)$ for $B_1$ and $B_2$, indicating that the q-resolution is weakly correlated to gun phase. The GP predicted PF curve (denoted with green squares) is plotted in Fig. 2b, c. The GP predicted PF curve connects the GP predicted optimal set points for lowest $\sigma_t$ at [$B_1 = 0.131$, $B_2 = 0.240$] and lowest $\sigma_q$ at [$B_1 = 0.123$, $B_2 = 0.217$]. Here, we define the empirical non-dominated data points as the approximate PF. The approximate PF points are plotted with green stars, the difference between approximate PF and GP predicted PF indicates that further exploration is needed to obtain the ultimate PF, and the lighter colored regions (both in $\sigma_t$ and $\sigma_q$ plots) are particularly valuable and remain to be explored. Fig. 2d, e show the calculated EHVI acquisition function projected onto the $\phi$ - $B_2$ and $B_1$ - $B_2$ space, the most valuable next observation point was found to be at [$\phi = 46.0$, $B_1 = 0.125$, $B_2 = 0.237$]. This is consistent with the expectation that the EHVI should suggest exploring points on the predicted PF

**Table 1 | Beamline input parameter ranges and kernel length scales obtained using GP regression, length scales are normalized to unit cube input space**

| Parameter | Range /Unit | $\mathcal{L}(\sigma_t)$ | $\mathcal{L}(\sigma_q)$ |
|---|---|---|---|
| Gun phase ($\phi$) | [35, 60] degree | 1.095 | 0.822 |
| Gun sol. strength($B_1$) | [0.11, 0.17] T | 0.765 | 0.077 |
| Micro-focus sol. strength($B_2$) | [0.15, 0.30] T | 0.852 | 0.215 |

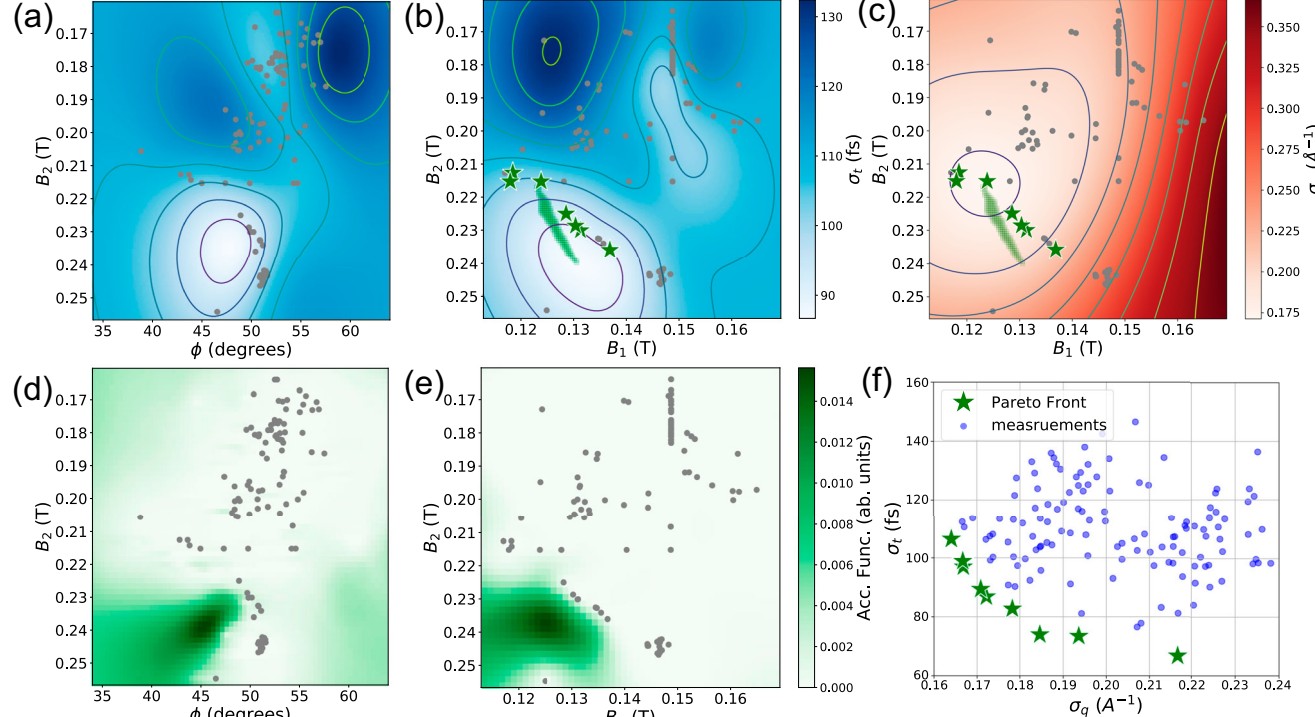

**Fig. 2 | Temporal length vs q-resolution optimization results. a**, **b** Overview of temporal length response projected to the $\phi$ - $B_2$ and $B_1$ - $B_2$ space. Gaussian Process surrogate model was constructed using measured data points(gray dots), color map(blue) represents the predicted mean of $\sigma_t$. **c** Overview of q-resolution response projected to the $B_1$ - $B_2$ space. The color map represents the predicted

mean of $\sigma_q$. **d**, **e** On-the-fly acquisition function projected to the $\phi$ - $B_2$ and $B_1$-$B_2$ space, colormap shows the optimal next observation regions predicted by the acquisition function. **f** Measured data points in the objective($\sigma_t$ - $\sigma_q$) space, approximate Pareto Front points are highlighted with green stars. The corresponding hypervolume convergence plot is shown in Supplementary Fig. 1.

curve to efficiently improve the hypervolume. Here, the advantage of the MOBO scheme can be clearly seen. It strategically proposes the next observation point which optimally increase the Pareto Frontier HV, and is thus more data efficient than a broad, undirected search for the PF. Fig. 2f shows the measured data points in the objective space. The approximate PF shows clear trade-off between $\sigma_t$ and $\sigma_q$. The smallest $\sigma_q$ was found to be 0.162 Å$^{-1}$ at $\sigma_t = 108$ fs, while smallest $\sigma_t = 67$ fs can be achieved with $\sigma_q = 0.217$ Å$^{-1}$.

We then used MOBO to conduct spot size ($\sigma_x$) vs $q$-resolution ($\sigma_q$) explorations by varying $B_1$ and $B_2$ at 10 fC, 50 fC and 100 fC initial pulse charges, the input parameter ranges used are shown in Supplementary Table 1. For each run, 10 random sampling points were used to initialize the exploration and in total 150 measurements were taken. Figure 3a illustrates the exploration results, all measured data points are plotted in the $\sigma_x$ - $\sigma_q$ space. The approximate PFs clearly shows trade-offs between $\sigma_x$ and $\sigma_q$ at different initial pulse charges. We observe that the approximate PF obtained at lower pulse charge (10 fC) shows significant improvement compared with higher pulse charge cases (50 fC and 100 fC). This observation is consistent with previous observations that smaller $\sigma_x$ or $\sigma_q$ can be achieved by lowering initial pulse charge, as spot size and q-resolution are limited by SC induced emittance growth at 4.2 MeV beam energy. At 10 fC initial pulse charge, the smallest $\sigma_x$ was found to be 45 $\mu$m at $\sigma_q = 0.351$ Å$^{-1}$, while smallest $\sigma_q = 0.162$ Å$^{-1}$ can be achieved with $\sigma_x = 100\,\mu$m. We also observe that $\sigma_q$ flattens with $\sigma_x > 100\,\mu$m, implying that other source of broadening on the diffraction detector, such as the detector Point Spread Function (PSF) are the primary limitation for further improvement in the $q$-resolution. We ran 9 independent optimizations for the 10 fC initial pulse charge case, each with a different set of 10 randomly sampled initial points. After each measurement, the hypervolume of the approximate PF point set was calculated in normalized objective space. Fig. 3b shows the convergence plot of approximate PF hypervolume versus number of iterations. One can clearly see improvement of hypervolume across the optimization process. The averaged hypervolume (blue) reaches 95% of its maximum within 30 measurements. We conducted a simulated Grid Search (GS) with step size equal to 10% of the parameter space subdomain using the interpolated dataset. The hypervolume obtained using GS was 62% of that obtained using MOBO after 30 measurements and 91% after 100 measurements. A direct comparison between MOBO and GS scheme is plotted in Fig. 3b, the advantage of MOBO to improve both search efficiency and maximum achievable hypervolume can be clearly seen.

## Discussion

Here we have experimentally demonstrated MOBO scheme to autonomously and efficiently optimize a subset of critical electron beam properties for an MeV-UED facility. The MOBO algorithm was used for sampling the multi-dimensional parameter space efficiently with little prior knowledge of the experimental system, which makes it practical for use in online beam tuning. By proposing solutions which optimally increase the hypervolume at each measurement step, the MOBO algorithm was capable to active-learn the PFs which give the trade-offs between key beam properties of interest for scientific experiments. The PF offers an unprecedented overview of the machine's performance limitations and can greatly assist human scientists in rapid decision-making, thereby maximizing the scientific output within the very limited beam time. The MOBO scheme was capable to achieve system performance comparable with that obtained by experienced operators and requires significantly fewer measurements compared with traditional exploration methods such as a GS scan. It is anticipated that this advancement will have a direct and substantial impact on the scientific experiments conducted at the SLAC MeV-UED facility, particularly in the studies which require simultaneously optimizing electron pulse length and q-resolution/spot size, such as the studies of charge density wave orders[31], ultrafast energy transfer in small-scale, low-dimensional systems[32], and other complex phenomena such as unconventional superconductivity in twisted-angle graphene super lattices[33]. While this work focuses on the application of the MOBO scheme in electron beam tuning for MeV-UED, this algorithm is flexible, efficient and can be used to replace GS in any experimental scenarios that require online explorations in a multi-dimensional, nonlinear parameter space while simultaneously optimizing multiple objectives, and we expect it to have broad impact across researches in different diciplines (physics, chemistry and biology) at large scale, complex scientific user facilities, such as free electron lasers, synchrotron radiation light sources and neutron sources.

While our demonstration of MOBO for online tuning was conducted in a relatively low-dimensional input parameter space, it is expected that MOBO can be scaled up to higher dimensional parameter space[34]. During our experiment, the extra computation time associated with GP fitting and EHVI acquisition function optimization is small relative to the reduction in electron beam property evaluation time associated with faster convergence of HV (as shown in Supplementary Fig. 2). One major challenge is the rapidly increasing computational cost when the number of data points becomes large, as is often the case when conducting explorations in high dimensional parameter spaces. One solution is to use Neural Network (NN) surrogate models as the prior mean to improve the BO efficiency, even with

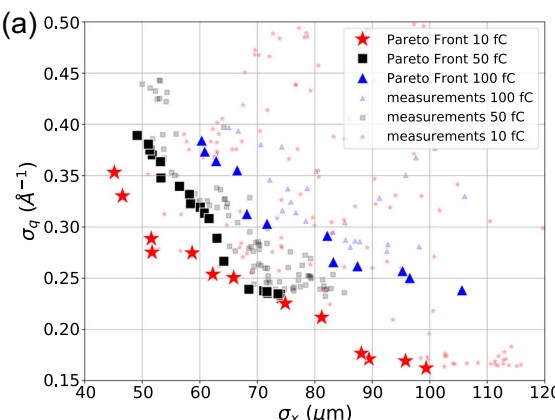
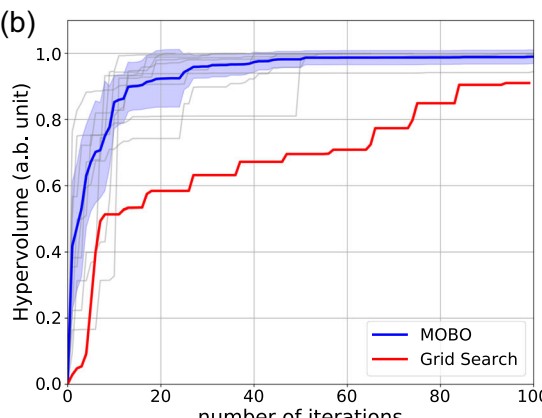

**Fig. 3 | Beam spot size vs q-resolution optimizations. a** Optimization results with 10 fC, 50 fC and 100 fC initial pulse charges. Measured data points are plotted in the $\sigma_x$-$\sigma_q$ space, approximate Pareto Front points are highlighted. **b** Convergence plot with averaging Pareto Frontier hypervolume (blue) for 9 independent optimizations (gray) for the 10 fC initial pulse charge case, shading area denotes 1 sigma variance. Interpolated grid search result (red) is plotted for comparison.

erroneous predictions[35]. Work in the multi-objective Bayesian optimization field has produced efficient methods for HV and acquisition function calculations[36]. Finally, if the evaluations can be conducted in parallel, as in simulations or under the multi-beamline scenario, recently developed parallelized acquisition functions, such as q-Expected Hypervolume Improvement (q-EHVI)[37] can be used to significantly reduce overall optimization time while maintaining the sampling efficiency advantages of EHVI.

## Methods

### Gaussian process modeling

Nonparametric GP surrogate models were used to predict the value of an objective function $f(\mathbf{x})$ using Bayesian statistics[24]. The GP model includes a distribution of possible functions $y(\mathbf{x}) \sim \mathcal{GP}(\mu(\mathbf{x}), k(\mathbf{x}, \mathbf{x}'))$. With $\mu(\mathbf{x})$ presenting the predicted mean of objective function $y(\mathbf{x})$, $k(\mathbf{x}, \mathbf{x}')$ is the kernel function based on the objective function behavior[20]. Given N previous measurements $\mathcal{D} = \{(x_1, y_1), (x_2, y_2), \ldots, (x_N, y_N)\}$, the predictive probability distribution of the function value is given by a normal distribution:

$$p(y|\mathcal{D}, \mathbf{x}) = \mathcal{N}(\mu(\mathbf{x}), \sigma^2(\mathbf{x})) \tag{1}$$

where

$$\mu(\mathbf{x}) = \mathbf{k}^T [\mathbf{K} + \sigma_n^2 \mathbf{I}]^{-1} \mathbf{y} \tag{2}$$

$$\sigma(\mathbf{x}) = k(\mathbf{x}, \mathbf{x}) - \mathbf{k}^T [\mathbf{K} + \sigma_n^2 \mathbf{I}]^{-1} \mathbf{k} \tag{3}$$

$$\mathbf{k} = [k(\mathbf{x}, \mathbf{x}_1), \ldots, k(\mathbf{x}, \mathbf{x}_N)]^T \tag{4}$$

$$\mathbf{K} = \begin{bmatrix} k(\mathbf{x}_1, \mathbf{x}_1) & \cdots & k(\mathbf{x}_1, \mathbf{x}_N) \\ \vdots & \ddots & \vdots \\ k(\mathbf{x}_N, \mathbf{x}_1) & \cdots & k(\mathbf{x}_N, \mathbf{x}_N) \end{bmatrix} \tag{5}$$

where $\mathbf{y} = [y_1, y_2, \ldots, y_N]^T$ and $\sigma_n$ represents the noise hyperparameter. In this work, a Matérn kernel function was used for GP modeling:

$$k_{\text{Matérn}}(\mathbf{x}, \mathbf{x}') = \frac{2^{1-\nu}}{\Gamma(\nu)} \left( \frac{\sqrt{2\nu} \, \|\mathbf{x} - \mathbf{x}'\|}{\mathcal{L}} \right)^{\nu} K_{\nu} \left( \frac{\sqrt{2\nu} \, \|\mathbf{x} - \mathbf{x}'\|}{\mathcal{L}} \right), \tag{6}$$

With $\|\mathbf{x} - \mathbf{x}'\|$ represnts the Euclidean distance between inputs, $\Gamma$ is the gamma function, and $K_\nu$ is the modified Bessel function. $\nu = 2.5$ was used in this work for modeling functions in the absence of prior information. The hyperparameter $\mathcal{L}$ determines the characteristic length-scale of the kernel, where small and large values correspond to rapidly and slowly varying functional behavior respectively.

### Multi-objective Bayesian optimization

The goal of multi-objective optimization is to search the input parameter space of the experimental system and to optimize the vector objective function $\mathbf{y}(\mathbf{x}) = \{y_1(\mathbf{x}), y_2(\mathbf{x}), \ldots, y_M(\mathbf{x})\}$. Usually, there is no best single solution $\mathbf{x}^*$ that can simultaneously optimize all objectives. Instead, multi-objective optimization algorithms attempt to find a set of points, known as the PF that optimally balances the trade-offs between the competing objectives. The Pareto Frontier set $\mathcal{P}$ is defined as the set of non-dominated points in objective space. For the minimization case, an objective vector $\mathbf{y}(\mathbf{x})$ dominates another vector $\mathbf{y}(\mathbf{x}')$ if $y_m(\mathbf{x}) \leq y_m(\mathbf{x}')$ for all $m = 1, \ldots, M$ and there exists at least one $n$ such that $y_n(\mathbf{x}) < y_n(\mathbf{x}')$. HV is an often used metric to evaluate the quality of a PF. The HV is defined as the M-dimensional Lebsegue measure of the subspace dominated by $\mathcal{P}$ and bounded below by a reference point $\mathbf{r}$[30]:

$$HV(\mathcal{P}) = \lambda_M(\cup_{\mathbf{y} \in \mathcal{P}} [\mathbf{y}, \mathbf{r}]) \tag{7}$$

Here $\lambda_M$ is the Lebesgue measure on $\mathbb{R}^M$. A larger HV corresponds to a better solution set $\mathcal{P}$. The MOBO technique aims to maximize the HV through the following process. Given a set of $N$ observations: $D_N = \{(\mathbf{x}_1, \mathbf{y}_1), (\mathbf{x}_2, \mathbf{y}_2), \ldots, (\mathbf{x}_N, \mathbf{y}_N)\}$, each objective is modeled as an independent GP surrogate model:

$$y_m(\mathbf{x}) \sim \mathcal{GP}_m[\mu_m(\mathbf{x}), k_m(\mathbf{x}, \mathbf{x}')] \tag{8}$$

The second part of MOBO is to construct an acquisition function to propose points which are likely to maximally increase the Pareto Frontier HV. A commonly used acquisition function is the Expected Hypervolume Improvement (EHVI)[30], which calculates the average increase in hypervolume using the posterior probablity distribution of each objective function from the surrogate model. The EHVI acquisition function is defined as:

$$\alpha_{EHVI}(\mu, \sigma, \mathcal{P}, \mathbf{r}) := \int_{\mathbb{R}^M} H_I(\mathcal{P}, \mathbf{y}, \mathbf{r}) \cdot \mathcal{N}_{\mu, \sigma}(\mathbf{y}) d\mathbf{y} \tag{9}$$

With $H_I(\mathcal{P}, \mathbf{y}, \mathbf{r})$ being the hypervolume improvement from an observed point $\mathbf{y}$ in objective space, $\mathcal{N}_{\mu, \sigma}$ is the multivariate normal distribution function defined by the GP predicted mean $\mu$ and uncertainty $\sigma$. The hypervolume improvement is defined by the exclusive HV contribution to the current PF by adding $\mathbf{y}$ to the Pareto set. Combining the GP surrogate model and acquisition function, MOBO optimization can be performed. In each iteration, MOBO evaluates a candidate point in the parameter space and updates the GP surrogate model with the newly acquired data. The EHVI acquisition function is then used to calculate the next observation point in the parameters space. The process is performed iteratively and ultimately providing an approximation of the best achievable PF.

### Determination of temporal length

The measurements of temporal length of electron pulses at IP was accomplished through interaction with quasi-single-cycle THz pulses generated via optical rectification of 800 nm laser pulses in a $LiNbO_3$ crystal[38,39]. An Off-Axis Parabolic (OAP) mirror was used to focus the THz pulses into a high field enhancement parallel-plate waveguide (PPWG)[40] streaking structure placed at IP. The THz pulses are then focused at the 70 μm slit in the PPWG and impart a transverse momentum kick to the electron beam which streaks the longitudinal profile onto the y axis on the downstream detector. The single-shot temporal resolution of THz streaking diagnostics was better than 1.5 fs with 3 μJ THz pulse energy.

### Determination of q-resolution

The q-resolution quantifies the extent to which tiny features can be resolved within a given diffraction pattern. The instrumental q-resolution can be expressed as[41]:

$$\sigma_q = \sqrt{\sigma_{ND}^2 + \sigma_E^2} \tag{10}$$

$\sigma_{ND}$ is the non-dispersive broadening which depends on transverse beam emittance and focusing solenoid settings. $\sigma_E$ is the dispersive component which depends on beam energy spread $\Delta E$ and scattering wave vector $q$:

$$\sigma_E = \frac{\Delta E}{E_k} \left( \frac{\gamma}{\gamma + 1} \right) q \tag{11}$$

$E_k$ and $\gamma$ are the beam kinetic energy and Lorentz factor. During experiment, $\sigma_{ND}$ was measured using the electron diffraction detector (phosphor screen coupled with an Electron-Multiplying CCD) located 3.2 m downstream the IP. Reciprocal space calibration was performed using a single crystal gold sample. Within the experimental parameter range, $\sigma_E$ is calculated to be $<0.011\,\text{Å}^{-1}$, thus the $q$-resolution is dominated by $\sigma_{ND}$.

## Data availability

The raw data generated in this study have been deposited in the Zenodo repository under accession code: https://doi.org/10.5281/zenodo.11095450[42].

## Code availability

This research used open source Python software libraries, including BoTorch[43] and Xopt[44]. The data analysis codes are available from the corresponding author upon request.

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

## Acknowledgements
This work was supported by U.S. Department of Energy Office of Science, Office of Basic Energy Sciences (Contract No. DE-AC02-76SF00515 and DE-SC0021984). M. M. acknowledges the support from DOE Fusion Energy Science under FWP No. 100182. The SLAC MeV-UED program is supported by the U.S. Department of Energy Office of Science, Office of Basic Energy Science under FWP 10075, 100713 and 100940.

## Author contributions
F.J., A.E. and R.J.E. designed the experiment. A.E., R.R., F.J. and C.M. wrote the software required to perform data collection and online multi-objective bayesian optimizations. F.J., A.E., X.S., S.M., S.W., D.L., M.M., A.R., M.A.K.O., P.K., X.W. and E.N. conducted the experiment. R.J.E. and M.M. provided guidance and oversights. F.J., R.J.E., X.S. and R.R. performed data analysis and wrote the manuscript. All authors contributed to the completion of the manuscript.

## Competing interests
The authors declare no competing interests.
