## [Peer Review File · Nature Communications]

Multi-Objective Bayesian Active Learning for MeV-ultrafast electron diffractionReviewer #1 (Remarks to the Author):

In this manuscript, Ji et al. proposed an efficient online parameter tuning method based on a multi-objective Bayesian optimization-based (MOBO) active learning strategy for the beam tuning task at the SLAC MeV-UED facility. By searching the optimal parameters including the temporal length, q-resolution and spot size at the sample position, and exploring the trade-offs between them, the authors obtained comparable parameters to those obtained by traditional methods but with much fewer trials or measurements. In my opinion, this work is an efficient and low-cost solution for the simultaneous optimization of multiple complementary restricted parameters in the SLAC MeV-UED facility, and is worth promoting to various high dimensional and correlated systems. Nevertheless, I wonder if it is still too specialized for such a multidisciplinary journal. My comments are listed below:

1. The authors mentioned that the MOBO scheme has been employed to maximize the performances of complex facilities, including LCLS, AWA and LPAs at least. In general, I think it is difficult to get the main innovation point of the MOBO method in this work compared with the previous ones.
2. In Fig. 2(b) and (c), the authors found difference between the approximated PF and GP predicted PF, but stopped further exploration to achieve an ultimate PF. What are the concerns? I suggest the authors add some more measurements or discussions.
3. In Fig. 3(a), I noticed that the measurement distribution of 10 fC initial pulse charge case is much larger than those of the 50 fC and 100 fC. What's the reason for this?
4. There are some typos in the manuscript, which should have been avoided.

Reviewer #2 (Remarks to the Author):

The authors have adeptly showcased their development of utilizing multi-objective Bayesian optimization (MOBO) for MeV-ultrafast electron diffraction measurements. Large-scale experiments involving multi-parameter optimization indeed present complexities. It's commendable that there are initiatives to incorporate advanced algorithms in this domain. The SLAC, with which the authors are associated, has a notable track record with several significant publications on this subject, including 'Bayesian Optimization of a Free-Electron Laser' in 2020 and 'Multiobjective Bayesian Optimization for online accelerator tuning' in 2021. Nevertheless, the current manuscript offers a fresh perspective on the application of MOBO. There's a distinct technical novelty in this development. The foundational concept behind the authors' work is undoubtedly sound, and they have presented the algorithm's diagram with clarity.

Given that MOBO is an extensively studied algorithm and is accessible through the open-source library BoTorch (<https://botorch.org/docs/introduction>), its stellar performance in MeV-ultrafast electron diffraction measurements is in line with expectations. Overall, the authors' work is robust. While some parts of the writing may benefit from further clarity, the manuscript holds considerable promise for publication in NC. Addressing certain points could enhance its potential further:

1. With a number of publications emerging from SLAC on the use of Bayesian Optimization for large-scale experiments, upon reading this manuscript, my initial anticipation was to discern the key distinctions from prior work. This aspect seems to be sparingly covered in the manuscript.
2. The realm of multi-objective optimization boasts numerous renowned algorithms, making it challenging to ascertain the superiority of one over the others. Bayesian optimizations are among the most sought-after methods in this field. The authors have adapted this well-established concept to the context of MeV-ultrafast electron diffraction measurement. This may be inherited by the previous success of other applications from SLAC. I can see some arguments from the authors for a special reason of only focusing on this direction. However, I would expect a reasonable study from the literature review to convince the readers of the unique advantage of MOBO.

3. Within the 'Results' section, the opening subsection titled 'Multi-Objective Bayesian Optimization' primarily recapitulates the MOBO algorithm as delineated in past literature. The presented style makes it a tad challenging to pinpoint the unique contributions made by the authors.

4. To my comprehension, Fig.1 seems to correlate with the 'Method' section. Yet, a thorough explication of the mathematical model appears to be missing in this section.

5. The authors have underscored the computational time efficiency of MOBO. However, the elucidation on this advantage seems brief, with a mere reference to 'an average of less than 5 s in each optimization step'. A more comprehensive presentation of these results would be beneficial.

Response to Referees

We sincerely thank the two referees for their comments and questions on the manuscript. Their comments helped us improving the paper and discussing crucial points in greater detail. In the following we report the point-by-point answers to the questions/comments raised.

Reviewer #1 (Remarks to the Author):

In this manuscript, Ji et al. proposed an efficient online parameter tuning method based on a multi-objective Bayesian optimization-based (MOBO) active learning strategy for the beam tuning task at the SLAC MeV-UED facility. By searching the optimal parameters including the temporal length, q-resolution and spot size at the sample position, and exploring the trade-offs between them, the authors obtained comparable parameters to those obtained by traditional methods but with much fewer trials or measurements. In my opinion, this work is an efficient and low-cost solution for the simultaneous optimization of multiple complementary restricted parameters in the SLAC MeV-UED facility, and is worth promoting to various high dimensional and correlated systems. Nevertheless, I wonder if it is still too specialized for such a multidisciplinary journal. My comments are listed below.

Response: We greatly appreciate the comments of the referee. In this work, we implemented the Multi-Objective Bayesian Optimization(MOBO) algorithm to enhance the operations of complex scientific user facilities(SUF) such as SLAC MeV-UED. To the best of our knowledge, this marks the first instance where MOBO has been applied to actively learn and navigate through the trade-offs of key beam properties (pulse length, spot size, and q-resolution) that have a direct impact on the outcome and quality of scientific user experiments. It is anticipated that this advancement will have a direct and substantial impact on the scientific experiments conducted at the SLAC MeV-UED facility, and we expect it to have broad impact across researches in different disciplines (physics, chemistry and biology) at large scale, complex SUFs. We have addressed all the comments, and the following are the detailed responses.

1. The authors mentioned that the MOBO scheme has been employed to maximize the

performances of complex facilities, including LCLS, AWA and LPAs at least. In general, I think it is difficult to get the main innovation point of the MOBO method in this work compared with the previous ones.

Response: In this work, we have implemented the MOBO algorithm to enhance the operations of complex SUFs such as SLAC MeV-UED. We'd like to emphasize that the LCLS [J. Duris et al, Phys. Rev. Lett, vol. 124, p. 124801 (2020)] and AWA [R. Roussel et al, Nature Communications 12, 5621 (2021)] works were focused on single objective optimization. While the LPA work [S. Jalas et al, Phys. Rev. Accel. Beams, vol. 26, p. 071302 (2023)] was aimed at demonstrating the MOBO scheme on a limited set of available test parameters, our work is aimed at using the MOBO optimization to directly improve operation of an active facility by focusing on the most important objectives of interest for user experiments. MeV-UED provides an ideal test case for this due to its being a relatively self-contained facility where these types of R&D studies are more easily decoupled from the broader facility operation.

One of the most significant advantage of MOBO is its high data efficiency comparing with other optimization techniques such as evolutionary algorithms. As a result, we were able to obtain the Pareto Front (PF) within 150 measurements in a relatively short period (currently approximately 3 hours). The PF offers an unprecedented overview of the machine's performance limitations. By providing human scientists with a machine-learned reference of the PF, it greatly assists in rapid decision-making to select desired working points, thereby maximizing the scientific output within the very limited beam time.

We anticipate that this advancement will have a direct and substantial impact on the scientific experiments conducted at the SLAC MeV-UED facility. It is expected to facilitate groundbreaking research into ultrafast structural dynamics, particularly in the study of charge density wave orders[A. Kogar, et al. Nat. Phys. 16, 159-163(2020)], ultrafast energy transfer in small-scale, low-dimensional systems[A. Sood, et al. Nat. Nano. 18, 29-35 (2023)], and other complex phenomena such as unconventional superconductivity in twisted-angle graphene super lattices [Y. Cao, et al. Nature 556, 43-50(2018)]. By enhancing the efficiency and effectiveness of these investigations, our application of MOBO holds the potential to unlock new understandings in these advanced areas of study.

To emphasize the main innovation point and impact of this work, we have added the following sentences in the first paragraph of the Discussion section:

“the MOBO algorithm was capable to active-learn the PFs which give the trade-offs between key beam properties (in our study electron pulse length, q-resolution and spot size at sample) of interest for scientific experiments. The PF offers an unprecedented overview of the machine’s performance limitations and can greatly assist human scientists in rapid decision-making, thereby maximizing the scientific output within the very limited beam time”

...

“It is anticipated that this advancement will have a direct and substantial impact on the scientific experiments conducted at the SLAC MeV-UED facility, particularly in the studies which require simultaneously optimizing electron pulse length and q-resolution/spot size, such as the studies of charge density wave orders [A. Kogar, et al. Nat. Phys. 16, 159-163(2020)], ultrafast energy transfer in small-scale, low-dimensional systems [A. Sood, et al. Nat. Nano. 18, 29-35 (2023)], and other complex phenomena such as unconventional super-conductivity in twisted-angle graphene super lattices [Y. Cao, et al. Nature 556, 43-50(2018)].”

2. In Fig. 2(b) and (c), the authors found difference between the approximated PF and GP predicted PF, but stopped further exploration to achieve an ultimate PF. What are the concerns? I suggest the authors add some more measurements or discussions.

Response: The referee is raising a crucial point. Indeed, the discrepancy between the approximated PF and the GP predicted PF suggests that additional exploration could lead to a more accurate ultimate PF.

In practice, due to the inherent complexity and uncertainties of the experimental system, the ultimate PF can never be fully achieved within a finite number of measurements. The GP model used for PF prediction is an approximation that relies on existing data. With more data, the model's accuracy could be improved, but this process is iterative and can be time-consuming. Figure R1 shows the HV convergence plot for the temporal length vs q-resolution optimization results. The HV start to converge with number of iterations larger than 90, and it clearly shows a

diminishing return for additional measurements. Given the limited availability of beam time, we aimed to demonstrate the potential of the MOBO approach within the constraints of a reasonably sized dataset.

Figure R1. HV convergence plot for the temporal length vs q-resolution optimization results.

At mean time, the calculated Expected Hypervolume Improvement (EHVI) acquisition function suggests that the next observations would be most valuable around specific regions, such as [$\varphi = 46.0$, $B1 = 0.125$, $B2 = 0.237$]. This is consistent with the expectation that the EHVI should suggest exploring points on the GP predicted PF curve to efficiently improve the hypervolume. This MOBO based strategic exploration is clearly more efficient than a broad, undirected search for the ultimate PF.

In light of the referee's suggestion, we recognize the importance of continued exploration in future works, these will include employing adaptive sampling techniques to improve the GP model's accuracy within a minimal number of additional measurements, as well as including physics informed GP models [A. Hanuka, et al. Phys. Rev. Accel. Beams 24, 072802 (2021)] for online optimizations. Currently, the primary limitation is the time-taking electron beam diagnostics (as discussed in response to reviewer 2, question 5). By implementing highly efficient single-shot, non-destructive and automated e-beam diagnostics, more data points could be obtained within a shorter time. This enhancement would enable the construction of a more accurate GP model, leading to a better approximation of the ultimate PF.

To comply with the referee, we included Fig. R1 into the supplementary information and added following discussion in the 2nd paragraph of the "Results" section:

“Here, the advantage of the MOBO scheme can be clearly seen. It strategically proposes the next observation point which optimally increase the Pareto Frontier HV, and is thus more data efficient than a broad, undirected search for the PF.”

3. In Fig. 3(a), I noticed that the measurement distribution of 10 fC initial pulse charge case is much larger than those of the 50 fC and 100 fC. What’s the reason for this?

Response: The broader measurement distribution for the 10 fC initial pulse charge case was due to the slightly larger B_2 range that was used for the 10 fC case, as shown in table R1.

Initial Pulse Charge (fC)	Range of B_1 (T)	Range of B_2 (T)	ϕ (deg)
10	[0.13, 0.16]	[0.255, 0.364]	55
50	[0.13, 0.16]	[0.255, 0.326]	55
100	[0.13, 0.16]	[0.255, 0.326]	55

Table R1. Parameter range used in spot size vs q-resolution optimizations.

Furthermore, when we constrain the data points for the 10 fC case to the same parameter range as that used for the 50 fC and 100 fC cases, the measurement distribution becomes consistent, as shown in Figure R2.

Figure R2. Measurement distribution for the 10 fC, 50 fC and 100 fC case constrained to the same parameter range.

We included the parameter ranges used under the σ_x vs σ_q optimization cases in the supplementary information for reference.

4. There are some typos in the manuscript, which should have been avoided.

Response: We thank the referee for pointing out this, we have made an effort to correct typographical errors.

Reviewer #2 (Remarks to the Author):

The authors have adeptly showcased their development of utilizing multi-objective Bayesian optimization (MOBO) for MeV-ultrafast electron diffraction measurements. Large-scale experiments involving multi-parameter optimization indeed present complexities. It's commendable that there are initiatives to incorporate advanced algorithms in this domain. The SLAC, with which the authors are associated, has a notable track record with several significant publications on this subject, including 'Bayesian Optimization of a Free-Electron Laser' in 2020 and 'Multiobjective Bayesian Optimization for online accelerator tuning' in 2021. Nevertheless, the current manuscript offers a fresh perspective on the application of MOBO. There's a distinct technical novelty in this development. The foundational concept behind the authors' work is undoubtedly sound, and they have presented the algorithm's diagram with clarity.

Given that MOBO is an extensively studied algorithm and is accessible through the open-source library BoTorch (<https://botorch.org/docs/introduction>), its stellar performance in MeV-ultrafast electron diffraction measurements is in line with expectations. Overall, the authors' work is robust. While some parts of the writing may benefit from further clarity, the manuscript holds considerable promise for publication in NC. Addressing certain points could enhance its potential further.

Response: We thank the referee for the thorough review of the manuscript and the encouraging comments. We have addressed all the comments, and the following are the detailed responses. The revisions made in our manuscript align with the referee's recommendations and significantly strengthen the quality and clarity of our work.

1. With a number of publications emerging from SLAC on the use of Bayesian Optimization for large-scale experiments, upon reading this manuscript, my initial anticipation was to discern the key distinctions from prior work. This aspect seems to be sparingly covered in the manuscript.

Response: We thank the referee for the suggestion. While the single-objective Bayesian Optimization techniques have been extensively used for large-scale facilities at SLAC, our work is aimed at using the MOBO optimization to directly improve operation of an active facility by focusing on the most important objectives of interest for user experiments. As discussed in response to reviewer 1 (question 1), it is also the first time where MOBO has been applied to actively learn the trade-offs of key beam properties (pulse length, spot size, and q-resolution) that have a direct impact on the outcome and quality of scientific user experiments.

To comply with the referee, we have added the following description in the last paragraph of the Introduction section:

“and actively learn the PF giving trade-offs of key beam properties (electron pulse length, q-resolution and spot size at sample) that have a direct impact on the outcome and quality of scientific user experiments.”

2. The realm of multi-objective optimization boasts numerous renowned algorithms, making it challenging to ascertain the superiority of one over the others. Bayesian optimizations are among the most sought-after methods in this field. The authors have adapted this well-established concept to the context of MeV-ultrafast electron diffraction measurement. This may be inherited by the previous success of other applications from SLAC. I can see some arguments from the authors for a special reason of only focusing on this direction. However, I would expect a

reasonable study from the literature review to convince the readers of the unique advantage of MOBO.

Response: We thank the referee for the suggestion. Indeed, the goal of multi-objective optimization is to obtain the PF which represents the system performance limit and gives the trade-offs between key objectives. In this domain, one of the most popular methods are the evolutionary algorithms for their simplicity in implementation and broad applicability [H. R. Maier, et al., Environmental modelling & software 114, 195 (2019)]. However, the evolutionary algorithms are extremely data inefficient as they require parallelized evaluation of large number of candidate solutions to converge to the PF. As a result, the evolutionary algorithms are nearly impossible to use for online accelerator tunings.

On the other hand, MOBO with the EHVI acquisition function, addresses some of these inefficiencies. The EHVI approach is designed to prioritize the selection of points that are more likely to maximally expand the hypervolume of the PF, whereas genetic algorithms select points only based on their optimality. Consequently, the MOBO is much more data efficient and can converge to the PF at least one order of magnitude faster than evolutionary algorithms. A major advantage of the MOBO scheme is that it can be conducted in a serialized manner when parallelized evaluation is not possible, and is thus a critical step toward online tunings of real accelerator systems [R. Roussel, et al., Phys. Rev. Accel. Beams, vol. 24, p. 062801 (2021)]. To comply with the Referee, we have added the following discussion in the 3rd paragraph in the Introduction section:

“The goal of multi-objective optimization is to determine the Pareto Front (PF) which represents the system performance limit and gives the trade-offs between key objectives. In this domain, one of the most popular methods are the evolutionary algorithms for their simplicity in implementation and broad applicability [H. R. Maier, et al.]. However, the evolutionary algorithms are extremely data inefficient as they require evaluation of large number of candidate solutions to converge to PF. As a result, the evolutionary algorithms are nearly impossible to use for online accelerator tunings.[R. Roussel, et al. arXiv: 2312.05667v1]”

3. Within the 'Results' section, the opening subsection titled 'Multi-Objective Bayesian Optimization' primarily recapitulates the MOBO algorithm as delineated in past literature. The presented style makes it a tad challenging to pinpoint the unique contributions made by the authors.

Response: We agree with the referee's suggestion, the main innovation point of this work is implementation of MOBO algorithm for online optimization of a scientific user facility to benefit scientific discovery. we have moved the 'Multi-Objective Bayesian Optimization' subsection to the "Methods" section and added the following description in the first paragraph of the "Results" section to keep the main text consistent:

"In each iteration, MOBO evaluates a candidate point in the parameter space and updates the GP surrogate model with the newly acquired data. The Expected Hypervolume Improvement (EHVI) [29] acquisition function is then used to calculate the optimal next observation point (see methods). The process is performed iteratively and ultimately providing an approximation of the best achievable PF."

4. To my comprehension, Fig.1 seems to correlate with the 'Method' section. Yet, a thorough explication of the mathematical model appears to be missing in this section.

Response: We agree with the referee's suggestion, to present the mathematical model explicitly, we have included a "Gaussian Process modeling" subsection into the "Methods" section and moved the 'Multi-Objective Bayesian Optimization' subsection to the "Methods" section.

5. The authors have underscored the computational time efficiency of MOBO. However, the elucidation on this advantage seems brief, with a mere reference to 'an average of less than 5 s in each optimization step'. A more comprehensive presentation of these results would be beneficial.

Response: The referee raised a crucial point. Indeed, the performance scaling plays a key role in deciding which algorithm to use for online accelerator optimizations. The three typical

components of the MOBO process are GP model fitting, acquisition function optimization and GP model evaluation (in our case is the Electron beam property measurements). The scaling of GP model fitting follows the $O(N^3)$ growth as a function of number of data points [R. Roussel, et al. arXiv: 2312.05667v1]. Under the bi-objective case, the EHVI acquisition function follows the $O(N \log N)$ growth [Yang, K., Emmerich, M., Deutz, A. et al. J Glob Optim 75, 3–34 (2019)]. The actual computation time also depends on the algorithm framework (in this work it's Botorch) and CPU/GPU used. Figure R3 shows the computation time per iteration on a single-CPU under the σ_t vs σ_q optimization experimental condition. With ≤ 500 data points, the GP fitting time per iteration is below 4 s and EHVI optimization time is below 1 s per iteration, and thus the total computation time is below 5 s per iteration.

Figure R3. Computation time per iteration during the MOBO process

Currently, the major limitation is the time taking electron beam diagnostics associated with physically moving stages to switch between THz timing diagnostics and YAG/phosphor screens. The average time taken to measure electron beam properties is 60 s (shown as red dashed line in Figure R3), and it takes ~ 3 h to obtain ~ 150 data points for a typical MOBO run. By implementing highly efficient single-shot, non-destructive and automated electron beam diagnostics, we expect $\geq 10^3$ data points could be obtained within a shorter time. This enhancement could improve the accuracy of GP and fully exploit the advantages of the MOBO algorithm.

To comply with the referee, we included Figure R3 into the supplementary information for reference and added following description the last paragraph in the Discussion Section:

...“, the extra computation time associated with GP fitting and EHVI acquisition function optimization is small relative to the reduction in electron beam property evaluation time associated with faster convergence of HV (as shown in Supplementary Fig. 2)”

Reviewer #1 (Remarks to the Author):

The authors have addressed most of my concerns and enhanced on the novelty of the work in the response and revised manuscript. Therefore, I agree on its publication in NC.

Minor issues to address:

I noticed that the authors included Fig. R1 into the supplementary information, however, it is not related to the manuscript in text, which may make readers confused.

I raised the issue about typos in the previous comment. Unfortunately, I can still see 'Table.', 'Figure.', unnecessary ') in the first paragraph of "Results" section, etc.

Reviewer #1 (Remarks on code availability):

The URL directs to an open source Python software library for this research.

Reviewer #2 (Remarks to the Author):

The authors have made thorough and positive responses to both my review and those from others, for which I am grateful. Their attention to detail in addressing my concerns and revising the document has left me with no outstanding questions regarding the manuscript. This development showed considerable success at SLAC, and while its current field of application may be narrow, it serves as an exemplary case of employing MOBO in a large user facility. The concept is innovative, and both the results and methodology are solid. The clarity of the writing and the availability of the code in a GitHub repository further enhance the manuscript's merits. Consequently, I endorse its publication, with a minor reservation: The response to my query regarding the mathematical model in the 'Method – Gaussian Process Modeling' section was met with very detailed formulas. I am uncertain if this level of detail aligns with the editorial standards of Nature Communication. I defer to the editor's judgment on this matter for the final decision.

Reviewer #2 (Remarks on code availability):

I checked the github repository and found that MOBO was included in the xopt framework. It was not there when the xopt was published. Unfortunately, I couldn't locate any testing scripts related to the manuscript there. Without these, it's challenging to provide feedback on the performance as it pertains to the study. Nonetheless, given SLAC's esteemed reputation, I am confident they possess the necessary code for their work and evaluations.

Response to Referees

We sincerely thank the two referees for their positive comments on the manuscript. Their comments helped us further improving the paper.

In the following we report the point-by-point answers to the comments raised.

Reviewer #1 (Remarks to the Author):

The authors have addressed most of my concerns and enhanced on the novelty of the work in the response and revised manuscript. Therefore, I agree on its publication in NC.

Response: We greatly appreciate the comments of the referee. We have made further edits according to the referee's suggestions, and the following are the responses.

Minor issues to address:

I noticed that the authors included Fig. R1 into the supplementary information, however, it is not related to the manuscript in text, which may make readers confused.

Response: We thank the referee for pointing out this, to avoid confusion, we added following description in the figure caption of Fig. 2:

“The corresponding hypervolume convergence plot is shown in Supplementary Fig. 1.”

I raised the issue about typos in the previous comment. Unfortunately, I can still see 'Table.', 'Figure.', unnecessary ') in the first paragraph of "Results" section, etc.

Response: We thank the referee for pointing out this, we have made further efforts to correct typographical errors.

Reviewer #2 (Remarks to the Author):

The authors have made thorough and positive responses to both my review and those from others, for which I am grateful. Their attention to detail in addressing my concerns and revising the document has left me with no outstanding questions regarding the manuscript. This development showed considerable success at SLAC, and while its current field of application may be narrow, it serves as an exemplary case of employing MOBO in a large user facility. The concept is innovative, and both the results and methodology are solid. The clarity of the writing and the availability of the code in a GitHub repository further enhance the manuscript's merits. Consequently, I endorse its publication.

Response: We sincerely thank the referee for the review of the revised manuscript and the positive comments.

a minor reservation: The response to my query regarding the mathematical model in the 'Method – Gaussian Process Modeling' section was met with very detailed formulas. I am uncertain if this level of detail aligns with the editorial standards of Nature Communication. I defer to the editor's judgment on this matter for the final decision.

Response: We thank the referee for pointing out this. We defer to the editor's judgment on this matter for the final revision.